# Body Mass Index and Risk for COVID-19-Related Hospitalization in Adults Aged 50 and Older in Europe

**DOI:** 10.3390/nu14194001

**Published:** 2022-09-27

**Authors:** Maika Ohno, Dagmar Dzúrová

**Affiliations:** Research Centre on Health, Quality of Life and Lifestyle in a Geodemographic and Socioeconomic Context (GeoQol), Department of Social Geography and Regional Development, Faculty of Science, Charles University, Albertov 6, 128 43 Prague, Czech Republic

**Keywords:** COVID-19, obesity, BMI, older adults, diabetes, comorbidity, European population

## Abstract

Higher body mass index (BMI) has been associated with a higher risk for severe COVID-19 outcomes. The aim of this study was to investigate associations among BMI, underlying health conditions and hospital admission as well as the effects of COVID-19 vaccines in adults aged 50 years and older in Europe using data from the Survey of Health, Ageing and Retirement in Europe (SHARE) which was collected from June to August 2021, shortly after the second wave of the COVID-19 pandemic occurred in Europe. Survey data totalling 1936 individuals were used for statistical analyses to calculate the likelihood of hospitalization due to COVID-19 infection in relation to BMI, sociodemographic factors, comorbidities and COVID vaccination status. Approximately 16% of individuals testing positive for COVID-19 were hospitalized for COVID-19, and over 75% of these hospitalized individuals were either overweight or obese. The likelihood of hospitalization for individuals with obesity was approximately 1.5 times (CI [1.05–2.05]) higher than those with a healthy weight (BMI = 18.5–24.9 kg/m^2^) after adjusting for BMI, sex and age. After adjusting for sociodemographic factors, vaccination and comorbidities, the likelihood of hospitalization for individuals with obesity was 1.34 times higher than those with a healthy weight (CI [0.94–1.90]). Vaccine uptake was lowest in individuals with obesity (BMI ≥ 30 kg/m^2^) in all age groups. Individuals who had not received a vaccine were 1.8 times more likely to be hospitalized (CI [1.34–2.30]). Across European regions, obesity is associated with higher odds of hospitalization, and vaccination may be effective to reduce these odds for older adults.

## 1. Introduction

The number of deaths attributable to high body mass index (BMI) has doubled globally since 1990, and obesity and overweight are amongst the world’s greatest health problems [1]. The prevalence of overweight and obesity is increasing rapidly around the world, especially in Europe. In May 2022, the World Health Organization (WHO) published an extensive report on the current status of obesity and its impact on health including its effect on SARS-CoV-2 (COVID-19) morbidity and mortality in European regions [2], and called for urgent action to tackle this obesity epidemic in the region. Almost 60% of adults and one in three children in Europe are either overweight or obese [2]. On the contrary, in Asia-Pacific regions, the average of overweight prevalence among adults in low-income countries was no more than 31 and 43% in higher income countries [3]. Prevalence of overweight and obesity in Japan was 27% with obesity prevalence only 4.2% [3]. According to Eurostat data released in June 2022, six out of the ten countries within European regions with the highest prevalence of overweight and obesity in adult populations were among the Central and Eastern European Countries (CEEC) [4].

Obesity is defined as an excessive accumulation of body fat and has been known as a high-risk factor for diseases including high cholesterol, type 2 diabetes mellitus, hypertension, musculoskeletal diseases, cardiovascular diseases such as coronary heart disease and peripheral arterial disease, and certain cancer [5,6,7,8,9]. Obesity also has an adverse effect on lung function of patients with chronic obstructive pulmonary disease (COPD) and asthma [10]. One of the most common comorbidities among patients who were hospitalized for COVID-19 infection was obesity [11,12,13,14]. The COVID-19 pandemic has adversely exacerbated underlying health conditions of individuals with obesity. Furthermore, due to impaired immune response and low-grade inflammation in individuals with obesity [15], it has been speculated that the effectiveness of COVID-19 vaccines may be compromised as is often seen with seasonal influenza vaccines [16].

In the two years since the WHO declared the outbreak of COVID-19 a pandemic in March 2020, studies have suggested that there are regional differences in the outcomes of COVID-19 infection and mortality. European countries have been severely hit by COVID-19 with the death toll amounting to over two million as of July 2022 [17]. The United States (the US) and the United Kingdom (the UK) are also among countries with the highest COVID-19 mortality rates [18,19]. However, the estimated excess mortality due to the COVID-19 pandemic was highest in Central and Eastern Europe [18]. By comparison, COVID-19 mortality rates in Japan have been one of the lowest among the Organisation for Economic Cooperation and Development (OECD) countries despite the largest elderly population which is considered as a high-risk group [20].

Various studies have indicated a linear increase in the risk of developing severe COVID-19 symptoms with an increase in BMI. BMI greater than 23 kg/m^2^ increased the risk of hospitalization linearly in a cohort study with 6.9 million people [21]. A recent meta-analysis including 3,550,997 individuals from 32 countries showed overweight (BMI = 25–29.9 kg/m^2^) was associated with a 19% increase in the odds of hospitalization, but the odds of hospitalization for obesity (BMI ≥ 30 kg/m^2^) jumped to 72% [22]. Clearly, severity of obesity further aggravated the risk of hospitalization. A cohort study with a community-dwelling sample of over 300,000 people in the UK reported that the risk of hospitalization was 1.7 times higher in obese stage 1 (BMI 30 kg/m^2^ to <35 kg/m^2^), but jumped to more than three times higher in obese stage 2 (≥35 kg/m^2^) [23]. Mortality rates also increased with increasing BMI [24,25,26].

Previous studies have shown that older adults with COVID-19 are at high risk of severe COVID-19 symptoms [27,28,29,30]. However, interaction with differences in BMI, underlying health conditions, vaccination status and sociodemographic factors are not clear in this population. Furthermore, although the Central and Eastern European Countries have higher mortality due to COVID-19 [18] and higher obesity prevalence, associations between hospitalization, obesity and vaccination in CEEC are not yet thoroughly investigated.

Therefore, the aims of this study are, first, to evaluate the association between BMI and hospitalization in older adults (aged 50 and older); second, to investigate effects of sociodemographic factors, vaccination and comorbidities on hospitalization in this population; and third, to compare these characteristics in CEEC with other European regions in SHARE study.

## 2. Materials and Methods

### 2.1. Study Design

This is a secondary analysis of data collected from the Survey of Health, Ageing and Retirement in Europe (SHARE) Wave 8 and Wave 9, COVID-19 Survey 2 [31]. SHARE started in 2004 to study the life of people aged 50 and older across 28 European countries and Israel, collecting longitudinal data every two years. Datasets employed in the present study were retrieved from an online data catalogue of the SHARE Research Data Centre. Historically, SHARE data were collected via face-to-face interviews, however, the outbreak of COVID-19 during Wave 8 forced the suspension of such interviews in March 2020. At this point, approximately 70 percent of longitudinal interviews across countries had been completed. Limited by the pandemic, Wave 8 transitioned to what was called the SHARE Corona Survey 1 and was conducted by telephone (Computer-Assisted Telephone Interview; CATI) from June to August 2020, except for Austria whose field work finished at the end of September 2020. This was followed by Wave 9 SHARE Corona Survey 2 conducted the following year from June to August 2021, also by CATI. Questions in both SHARE Corona Surveys were limited to specific questions related to COVID infections and life changes during the on-going COVID-19 pandemic.

### 2.2. Study Sample

The present study focused on respondents who participated in Wave 8 and Wave 9 SHARE Corona Survey 2, aged 50 years and older, with no missing values for the variables selected in the analyses. Wave 8 was only used for information on education and BMI, which were not available in Wave 9 SHARE Corona Survey 2. Of the total 49,253 respondents, 18,239 were excluded for at least one of the following reasons: (1) aged 49 years or younger; (2) did not participate in the regular face-to-face Wave 8 and SHARE Corona Survey 2; (3) no available data for BMI; (4) no marital status information; (5) no education information; (6) no vaccination information; and (7) no hospitalization information.

The remaining 31,014 respondents were further reviewed for a COVID-19 test question. Since the aims of the present study were to examine associations between BMI and hospitalization for COVID-19 infection, we excluded respondents who never took a COVID-19 test before the Corona Survey 2 (June–August 2021) and respondents who never had a positive test result after Corona Survey 1, which ended August 2020. In other words, individuals who had a positive COVID-19 test result after Corona Survey 1 were included. This yielded data of 1936 respondents (‘study sample’) who were aged 50 and older with the selected variables and were included in the final analyses (Figure 1).

### 2.3. Observation Period

Data collection of Wave 9 SHARE Corona Survey 2 took place in European countries between 3 June and 14 August 2021. The exact dates of the survey in each European country are described in the study [32]. Figure 2 shows the duration of the survey period indicated by red vertical lines segmenting the pandemic waves experienced in Europe, and inverted triangles on the red lines indicate the start and end of the SHARE Corona Survey 2. The red-shaded zone indicates the time period corresponding to retrospective questions in Survey 2 while the green-shaded zone indicates that of Survey 1. The inverted triangles on the wave itself indicate the various peaks noting the average daily number of confirmed COVID-19 deaths per million people. The survey was conducted shortly after a large wave of pandemic caused deaths, known as the second wave, which swept across Europe from the end of November 2020 through April 2021, which resulted in over five thousand COVID-19-related deaths per day at its peak (approximately 7 deaths per million people) [33]. The COVID-19 vaccine rollout began gradually in each country starting in early 2021, and older adults were mostly one of the first groups to be vaccinated [34].

### 2.4. Measurements

The primary outcome examined in the present study was hospitalization among respondents who had tested positive for COVID-19. A total of eight covariates, four sociodemographic variables (Age, Sex, Education and European regions) and four health-related variables (BMI, Vaccination, Chronic lung disease, and Diabetes or high blood sugar), were included in binary logistic regression. Classification of BMI categories, age, education and CEEC are explained below.

BMI categories and obesity identification:

Information on BMI (kg/m^2^) was retrieved from Wave 8 and categorized into four standard categories of the WHO classification: Underweight (<18.5 kg/m^2^), Healthy weight (18.5 to 24.9 kg/m^2^), Overweight (25 to 29.9 kg/m^2^), and obesity (≥30 kg/m^2^).

Age:

Ages of study sample were categorized by 10-year subgroups: 50 to 59 years; 60 to 69 years; 70 to 79 years; and 80 years and older.

Education:

Based on ISCED-97 (International Standard Classification of Education) codes obtained from Wave 8 SHARE data, the education levels were categorized into four groups: None and Primary, Secondary, Post-secondary, and Tertiary education.

The Central and Eastern European Countries (CEEC):

European regions from the SHARE study were dichotomised into two groups: CEEC and All Other Countries. Countries included in CEEC were Bulgaria, Czechia, Estonia, Hungary, Lithuania, Latvia, Poland, Romania, Slovenia and Slovakia. All other countries included: Austria, Belgium, Croatia, Cyprus, Denmark, Finland, France, Germany, Greece, Israel, Italy, Luxembourg, Malta, The Netherlands, Portugal, Spain, Sweden and Switzerland.

Selection of underlying health conditions:

BMI, diabetes or high blood sugar, and chronic lung disease were selected for the analyses as potential risks for hospitalization as a result of COVID-19 since the growing number of studies have reported that individuals with a higher BMI, diabetes and/or chronic lung disease may have a higher risk of a severe course of COVID-19 [19,35,36].

### 2.5. Statistical Analysis

Statistical analyses were performed using IBM SPSS, Version 28 (IBM: Armonk, NY, USA). A descriptive analysis was performed to describe characteristics of the study sample of respondents who tested positive for COVID-19 after the SHARE Corona Survey 1 concluded in 2020, and the subsample who were hospitalized for COVID-19 among the study sample. The descriptive analysis included percentages for categorical variables, and mean and standard deviation for continuous variables.

Binary logistic regression was performed to examine associations between BMI, sociodemographic variables (sex, age and country) and underlying health conditions with hospitalization as a result of COVID-19. Binary logistic regression in the present study included three models, with a binary outcome variable being hospital admission (Yes, hospitalized = 1; Not hospitalized = 0).

Establishing Model 1, odds ratios (ORs) with 95% Confidence Intervals (CIs) were estimated between individuals with a positive COVID-19 test who were hospitalized for COVID-19 and those who were not hospitalized. For each model, explanatory variables were added. ORs were first adjusted for BMI (Reference = Healthy weight), sex (Reference = Female) and age (Reference = Age group 50–59).

Model 2 was fitted with sociodemographic predictors, i.e., education (Reference = Tertiary), and European regions (Reference = All other countries than CEEC) in addition to all the predictors (BMI, sex and age) in Model 1.

Adjusted for predictors in Model 1 and 2, Model 3 was further fitted with health-related predictors: vaccination status (Reference = Vaccinated) and underlying health conditions (Reference = No diabetes/high blood sugar and No chronic lung disease). To assess risks for hospitalization associated with COVID-19 in individuals, dichotomous variables were used for underlying health conditions (diabetes and chronic lung disease), COVID-19 vaccination status and hospitalization due to COVID-19.

All methods were carried out in accordance with relevant guidelines and regulations; no experiments on humans were done, and no human tissue samples or data were used. The datasets analysed in the study are available in http://www.share-project.org/special-data-sets/share-corona-survey-2.html. Ethical approval was not required for this secondary data analysis.

## 3. Results

### 3.1. Description of the Study Sample

Descriptive characteristics of the study sample (N = 1936 with 792 males (40.9%) and 1144 females (59.1%)) by COVID-19 and hospitalization due to COVID-19 are presented in Table 1. A total of 1936 respondents aged 50 and older, with a positive COVID-19 test result after July 2020 and with available data for BMI, sociodemographic variables (sex, age, education and country) and covariates were included in this study. Of these, nearly 16% were hospitalized with COVID-19.

There were more females (59.1%) who tested positive in the study sample as compared to 40.9% of males. However, slightly more male respondents (50.5%) were hospitalized with COVID-19. The mean age of the study sample was 68.9 ± 8.7 years, and 41.7% of the study sample were aged 70 years and older. The mean BMI of the study sample was 28.0 ± 4.9 kg/m^2^, which is classified as ‘overweight’ according to WHO classification. Approximately 1% of the study sample were underweight, and over 70% was either overweight or obese. Compared with this study sample, the subsample of individuals who had a positive COVID-19 test and were hospitalized for COVID-19 (mean age 72.4 ± 9.6 years) had a higher crude prevalence of obesity (34.5 versus 30.1%) and lower crude prevalence of healthy weight (24.1 versus 28.0%). Nearly 65% of the study sample had been vaccinated at least once before the survey period (June–August 2021, Wave 9 SHARE Corona Survey 2). However, vaccination uptake was lower among those who were hospitalized as compared to those who were not hospitalized (57 vs. 43%).

Figure 3 presents a relationship between hospitalization due to COVID-19 and vaccination status by BMI and age groups. A linear increase in hospitalization was observed with increasing age and BMI. Vaccination uptake was lowest among individuals with obesity in every age group. Vaccination rates for underweight in Age 50–59 and Age 80+ were 100%. The total number of underweight individuals was small. There was only one underweight individual in Age 50–59 and four underweight individuals in Age 80+, and they were all vaccinated. Approximately 60% of the study sample had secondary school education as compared to 57.3% for the subsample. While 13% of the study sample had no or primary school education, nearly 22% of the subsample had no or primary education.

### 3.2. Hospitalization Associated with COVID-19 Infection, BMI, and Other Covariates

Table 2 shows the outcomes of binary logistic regression, which includes three models designed to investigate associations of hospitalization due to COVID-19 among subjects (dependent variable) who had the COVID-19 test positive with covariates.

In Model 1, adjusted for a BMI, sex and age, the likelihood of hospitalization for individuals who were obese (BMI ≥ 30 kg/m^2^) was approximately 1.5 times higher (CI [1.05–2.05]) than those with a healthy weight (BMI = 18.5–24.9 kg/m^2^). The likelihood of hospitalization further increased significantly with age. Individuals who were aged 80 and older were four times more likely to be hospitalized (CI [2.48, 6.62]) as compared to younger individuals who were aged between 50 and 59 while the likelihood of hospitalization for individuals who were aged between 70 and 79 was twice as high (CI [1.34, 3.29]). The likelihood of being hospitalized for male was 1.5 times higher as compared to female (CI [1.18, 1.94]). After simultaneously controlling for three covariates (BMI, sex and age), the outcomes from the Model 1 indicate that for those who have tested positive for COVID-19, obesity, age 70 years old and above and being male are significant predictors of hospital admission associated with COVID-19.

Model 2 was extended by two additional covariates, i.e., the education level and the geographic variable (CEEC or All Other Countries) compared to Model 1. Obesity remained a significant predictor of hospitalization, and individuals with obesity were 1.4 times more likely to be hospitalized (CI [1.02, 2.01]) as compared to individuals with healthy BMI. The likelihood of hospitalization increased with age. Individuals aged 70 years and above were approximately twice as likely to be hospitalized as individuals aged 50–59 years (CI [1.24, 3.07]). Individuals aged 80 and older were 3.4 times more likely to be hospitalized (CI [2.05, 5.63]). The education level was confirmed as a significant covariate to the risk of hospitalization. The likelihood of hospitalization were approximately 1.9 times higher (CI [1.24, 3.07]) in the lower education group (None and primary) as compared to the tertiary education group. Further adjustment for CEEC did not alter the association.

Model 3 is a full model with all eight covariates considered. Three health-related variables (COVID-19 vaccination, chronic lung disease and diabetes or high blood sugar) were added to the variables from Model 2. COVID-19 vaccination and underlying health conditions were further added to Model 2 to examine the effect of vaccination and underlying health conditions known to be potential risk factors for hospitalization associated with COVID-19. Hence, chronic lung disease and diabetes/high blood sugar were included in the model. Model 3 shows that individuals who had not been vaccinated before the survey period were approximately 1.8 times more likely to be hospitalized (CI [1.34, 2.30]) as compared to those who had been vaccinated. With regard to underlying medical conditions, chronic lung disease indicated 3 times higher the odds of hospitalization (CI [2.14, 4.40]) as compared to those without the disease. The likelihood of being hospitalized for diabetes including high blood sugar were 1.4 times higher (CI [1.02, 1.93]) than those without diabetes. Controlling for vaccination status and underlying medical conditions slightly increased the likelihood of hospitalization for men with the odds ratio of 1.61 (CI [1.24, 2.08]) in Model 3. Though age and education also remained significant predictors of hospitalization, this additional control slightly reduced the ORs for age groups 70–79 by approximately 5% and for 80 and older by approximately 3%, as well as for the lower education group (none and primary) by approximately 4% as compared to Model 2. After additionally controlling for vaccination and underlying medical conditions, a reduction in the ORs for obesity was observed, while the ORs for obesity remained higher than the healthy weight though statistically non-significant (CI [0.94–1.9]).

## 4. Discussion

To our knowledge, this study is the first report to examine specifically the associations of BMI and other risk factors with hospitalization and the effect of vaccination on hospitalization among older adults aged 50 years and older across European regions and compared with CEEC. The survey was carried out across European regions between June and August 2021, not long after the worst pandemic wave swept across the regions, resulting in the highest death rates to date, after which by December 2020, the rollout of the first vaccine started in the EU.

The present study showed that over 75% of older adults who were hospitalized for COVID-19 were either overweight (BMI = 25–29.9 kg/m^2^) or obese (BMI ≥ 30 kg/m^2^), and approximately 35% of these hospitalizations were among the obese. In all our models the likelihood of hospital admission due to COVID-19 increased with higher BMI. An association between obesity and hospital admission remained significant after adjusting for sociodemographic covariates, i.e., sex, age, education and geographical region. This finding, in which obesity increases the likelihood of hospitalization as compared to healthy weight in adult population, is consistent with previous studies which suggested obesity was a strong predictor of risk of hospital admission [13,23,37,38]. In the US, 30% of 906,849 COVID-19-related hospital admissions were attributable to obesity as of 18 November 2019 [38]. In a large-scale UK cohort study with a sample of 334,329, it was reported that the associations with obesity and hospitalization were not affected after adjusting for sex, age and comorbidities, i.e., diabetes and hypertension [23]. Although the comorbidities, diabetes and hypertension, have been identified as risk factors of hospitalization and developing severe illnesses from COVID-19 [21,39,40,41], diabetes and hypertension were not found to be significant exposures in their model. The present study, however, comorbidities, i.e., diabetes or high blood sugar and chronic lung disease were shown to be a strong predictor for hospitalization due to COVID-19.

The present study confirmed findings from earlier studies that suggested male and older adults may be independent risk factors of hospital and/or ICU admission [37,42,43,44]. Consistent with a recent meta-analysis of 11,550 individual records [44], the present study revealed that individuals aged 70 and above and male remain significantly associated with hospitalization.

The protective effect of vaccines against hospitalization due to COVID-19 among people with obesity has been equivocal as compared to the protective effect for healthy weight. Obesity has been known to cause low-grade inflammation and impaired metabolic immune function, resulting in increased susceptibility to infectious diseases, for example, seasonal influenza [45]. Studies have indicated aggravation of severity of influenza symptoms due to impaired wound healing in lungs of obese mice [46] and decreased interferon, allowing replication of more virulent viruses [47]. Reduction in effectiveness of influenza vaccines has been observed among people with obesity as compared to people with healthy weight [16]. Reduced levels of anti-SARS-CoV-2 IgG has been observed with an increase in BMI in men who had received two doses of COVID-19 vaccines [48,49]. The present study, however, showed that individuals who had not been vaccinated were approximately 1.8 times more likely to be hospitalized due to COVID-19 than those who had been vaccinated. The ORs for obesity in our full model remained higher than that of healthy weight, but lower as compared to the ORs in Model 1 and Model 2. Similarly, the protective effect of vaccination among individuals with obesity was demonstrated in a recent population-based UK cohort study (91,1524 participants) which evaluated effectiveness of vaccination in relation to BMI and the risk of COVID-19 outcomes including hospitalization [39]. This study indicated that the ORs for hospitalization were reduced in all BMI categories after the first vaccination as compared to those without vaccination. The effectiveness of vaccines may not be the same in obesity as compared to healthy weight, however, this does not imply that vaccines will not protect individuals with obesity from hospital admission due to COVID-19. In the present study, we can assume that the vaccination respondents had received was the first dose given the fact that the survey was carried out before the second vaccine rollout which was at least 6 months after the first dose. This suggests that even one dose of vaccine may have a protective effect against hospitalization.

The present study showed that the vaccination uptake was lowest among individuals with obesity in every age group, however, factors that may have caused the lower vaccine uptake in people with obesity were not known from the survey. Recent studies have reported that higher COVID-19 vaccine hesitancy was observed among individuals with obesity [50,51]. These studies showed that fear and skepticism towards the COVID-19 vaccination increased with weight and were associated with higher vaccine refusal. Higher vaccine hesitancy was observed in certain sociodemographic factors such as religion, lower education attainment, ethnic groups, lower socioeconomic groups and migrant populations [52,53]. The association between higher prevalence of obesity and economically deprived groups of people has been known [2,54], and therefore, health inequality may be a contributing factor for lower COVID-19 vaccine uptake.

In comparison, people with obesity had the highest vaccine uptake in the UK [39]. This indicates that there are various factors that can affect vaccine uptake. Public policy may influence vaccine uptake. For example, the higher vaccination rate among people with obesity in the UK may be due to the fact that the UK COVID vaccination policy prioritized people with a BMI of 40 kg/m^2^ or higher [55]. Further, community engagement and efforts in building trust in vaccine efficacy resulted in higher vaccine uptake in some areas with minority ethnic groups in the UK [56]. Accessibility of vaccination centres might have affected vaccine uptake in older adults with obesity. Accessibility of vaccination centres is crucial for older adults with or without comorbidity. Geographical proximity and availability of vaccination centres depend on national vaccination strategies in each country [34]. Further research should investigate the causes of lower vaccination rates among older individuals with obesity who are at higher risks of severe COVID-19 symptoms.

Despite the higher prevalence of overweight and obesity in CEEC compared with other European countries reported in Eurostat, further adjustment for CEEC did not alter the association between obesity, male, older people 70 and above with hospitalization.

## 5. Limitations

The present study has some limitations. First, the present study is based on a self-reported survey, and information on individual health conditions, vaccination status and hospital admission associated with COVID-19 cannot be confirmed by health authorities. The number of vaccinations that respondents had received was not in the questionnaire, and therefore, it was assumed based on the EU vaccination deployment plans that most of respondents in the survey received no more than one vaccine. Second, the individual BMI information used in the present study was recorded in the previous year in Wave 8 which was suspended in March 2020 due to the COVID-19 pandemic, and it may not reflect actual BMI at the time of Wave 9 Corona Survey 2. However, a comparison of body weight of 19 million adults prior to the pandemic and during the pandemic showed a small or no weight change [57]. Furthermore, the above-mentioned population-based cohort study with 911,524 participants demonstrated that the study on vaccine effectiveness and risk of outcomes after vaccination was seen robust when using BMI measured within 2 years from the start of their study [39].

## 6. Conclusions

This study demonstrated the prevalence of overweight and obesity in older people aged 50 and above in that 75% were hospitalized, and individuals with obesity had a higher likelihood of hospital admission than with a healthy weight. The association with obesity and hospital admission may be reduced in vaccinated individuals. As vaccine uptake declined with increasing age and obesity, future research on underlying causes of lower vaccine uptake is needed. Given the fact that older adults are at higher risk of severe COVID-19 outcomes and European regions have higher obesity prevalence, we believe our findings in this study can contribute to future policy making.

## Figures and Tables

**Figure 1 nutrients-14-04001-f001:**
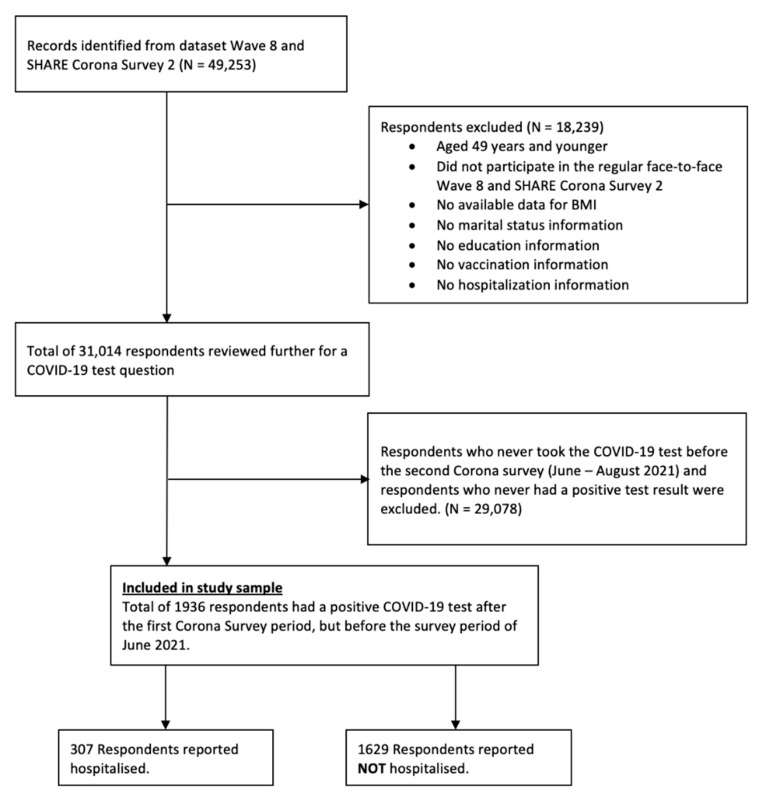
Flowchart showing selection of study sample.

**Figure 2 nutrients-14-04001-f002:**
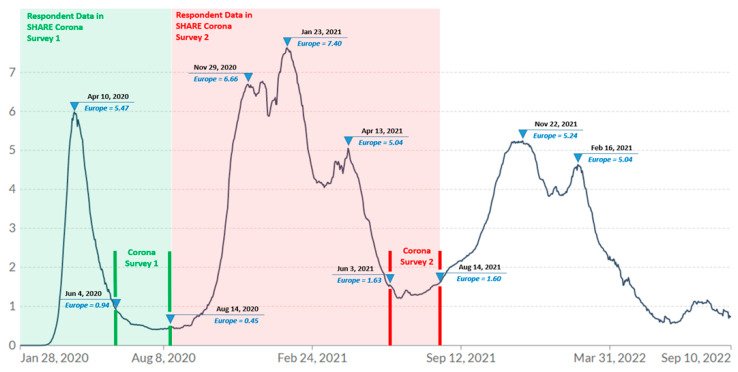
Daily new confirmed COVID-19 deaths per million people (7-day rolling average). Data from the COVID-19 Data Repository by the Center for Systems Science and Engineering (CSSE) at Johns Hopkins University via Our World in Data [33] with own modification indicating the survey period. https://github.com/CSSEGISandData/COVID-19 (accessed on 30 July 2022). Note: Inverted triangles on the red lines indicate the start and end of the SHARE Corona survey 2. The inverted triangles on the curve indicate the peaks of the pandemic waves with the average daily number of confirmed COVID-19 deaths per million people. The red-shaded zone indicates the time period corresponding to retrospective questions in Survey 2 while the green-shaded zone indicates that of Survey 1.

**Figure 3 nutrients-14-04001-f003:**
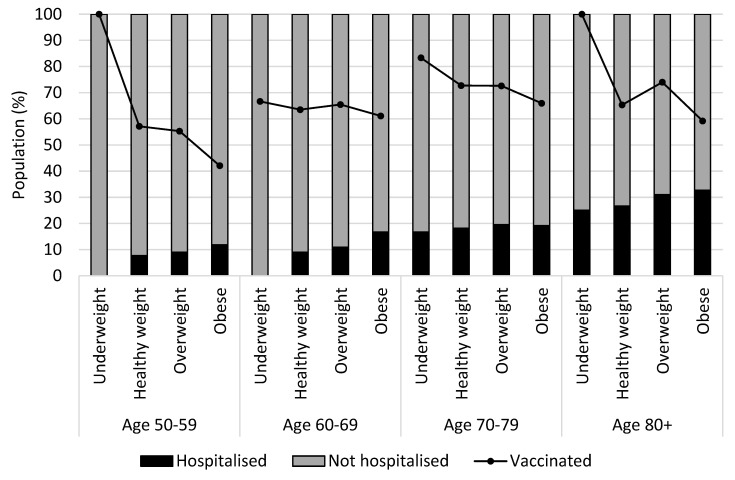
Hospitalization and vaccination status by BMI and age groups (*N* = 1936).

**Table 1 nutrients-14-04001-t001:** Descriptive characteristics of the study sample by COVID-19 test and hospitalization.

	Study Sample in the Analysis			
	Tested Positive for COVID-19 (*N* = 1936)		Hospitalized for COVID-19 (*N* = 307)	
	** *N* **	**(%)**	** *N* **	**(%)**
**Not Hospitalized**	1629	(84.1)	-	-
**Hospitalized**	307	(15.9)	-	-
**BMI categories (kg/m^2^)**				
Underweight (<18.5)	17	(0.9)	2	(0.7)
Healthy weight (18.5–24.9)	542	(28.0)	74	(24.1)
Overweight (25–29.9)	794	(41.0)	125	(40.7)
Obese (≥30)	583	(30.1)	106	(34.5)
**BMI continuous variable (kg/m^2^)**	**Mean**	**±SD**	**Mean**	**±SD**
	28.00	±4.90	28.82	±5.68
**Sex**				
Male	792	(40.9)	155	(50.5)
Female	1144	(59.1)	152	(49.5)
**Age group (years)**				
50–59	291	(15.0)	27	(8.8)
60–69	839	(43.3)	102	(33.2)
70–79	578	(29.9)	110	(35.8)
80+	228	(11.8)	68	(22.1)
**Age, years**	**Mean**	**±SD**	**Mean**	**±SD**
	68.88	±8.69	72.38	±9.60
**Education level**				
None and primary	251	(13.0)	66	(21.5)
Secondary	1168	(60.3)	176	(57.3)
Post-secondary	124	(6.4)	17	(5.5)
Tertiary	393	(20.3)	48	(15.6)
**European Regions**				
CEEC	1069	(55.2)	163	(53.1)
All the other countries	867	(44.8)	144	(46.9)
**Partner in household**				
Yes	1401	(72.4)	218	(71.0)
No	535	(27.6)	89	(29.0)
**COVID-19 Vaccination**				
Yes	1250	(64.6)	175	(57.0)
No	686	(35.4)	132	(43.0)
**Comorbidity**				
**Chronic lung disease**				
Yes	169	(8.8)	58	(18.9)
No	1762	(91.2)	249	(81.1)
**Diabetes or high blood sugar**				
Yes	320	(16.6)	73	(23.9)
No	1613	(83.4)	233	(76.1)
		(100)		(100)

**Table 2 nutrients-14-04001-t002:** Adjusted ORs and 95% CIs of hospitalization due to COVID-19 infection (Study sample, *N* = 1936).

	Model 1 ^†^			Model 2 ^‡^			Model 3 ^§^		
	OR	(95% CI)		OR	(95% CI)		OR	(95% CI)	
**BMI**									
Healthy (Ref)	1			1			1		
Underweight	0.72	(0.16, 3.31)		0.74	(0.16, 3.39)		0.61	(0.13, 2.94)	
Overweight	1.16	(0.84, 1.60)		1.16	(0.84, 1.60)		1.18	(0.85, 1.64)	
Obese	1.47	(1.05, 2.05)		1.43	(1.02, 2.01)		1.34	(0.94, 1.90)	
**Sex**									
Female (Ref)	1			1			1		
Male	1.51	(1.18, 1.94)		1.58	(1.23, 2.04)		1.61	(1.24, 2.08)	
**Age group**									
50–59 (Ref)	1			1			1		
60–69	1.26	(0.80, 1.97)		1.23	(0.78, 1.93)		1.20	(0.76, 1.89)	
70–79	2.10	(1.34, 3.29)		1.95	(1.24, 3.07)		1.85	(1.16, 2.95)	
80+	4.05	(2.48, 6.62)		3.40	(2.05, 5.63)		3.31	(1.97, 5.54)	
**Education**									
Tertiary (Ref)				1			1		
None and primary				1.87	(1.20, 2.90)		1.79	(1.14, 2.80)	
Secondary				1.14	(0.80, 1.62)		1.07	(0.75, 1.54)	
Post-secondary				1.07	(0.59, 1.97)		1.06	(0.57, 1.98)	
**European Regions**									
All other countries (Ref)				1			1		
CEEC				1.02	(0.78, 1.33)		0.87	(0.66, 1.15)	
**COVID-19 Vaccination**									
YES (Ref)							1		
No							1.75	(1.34, 2.30)	
**Diseases**									
**Chronic lung disease**									
No (Ref)							1		
Yes							3.06	(2.14, 4.40)	
**Diabetes**									
No (Ref)							1		
Yes							1.40	(1.02, 1.93)	

Notes: ^†^ Model 1: Adjusted for sex and age and BMI. ^‡^ Model 2: Adjusted for sex, age, BMI, education and European regions. ^§^ Model 3: Adjusted for sex, age, BMI, education, European regions, vaccination and diseases. The dependent variable: Hospitalization due to COVID-19, YES = 1, NO = 0.

## Data Availability

This paper uses data from SHARE waves 8 and 9 (DOIs: 10.6103/SHARE.w8.800 and 10.6103/SHARE.w9ca800). Börsch-Supan, A. (2022). Survey of Health, Ageing and Retirement in Europe (SHARE) Wave 8. Release version: 8.0.0. SHARE-ERIC. Data set. DOI: 10.6103/SHARE.w8.800. Börsch-Supan, A. (2022). Survey of Health, Ageing and Retirement in Europe (SHARE) Wave 9. COVID-19 Survey 2. Release version: 8.0.0. SHARE-ERIC. Data set. DOI: 10.6103/SHARE.w9ca.800.

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
