# Peer review of "Body Mass Index and Risk for COVID-19-Related Hospitalization in Adults Aged 50 and Older in Europe"

_nutrients, 2022, doi:10.3390/nu14194001_

Round 1

Reviewer 1 Report

The manuscript by Ohno et al reports on the relationship between BMI and hospitalization risk in a population of over 50 of central and eastern European countries.

The manuscript deals with a very important and current clinical issue, is well written but in my opinion requires some modification in the form and in the analysis of previous literature.

1. The period chosen by authors for analyses (2021 from June 15 to August 15) is the one with the lower incidence of COVID over the 3 years pandemic. Was this dome (chosen) on purpose? It would be very interesting to compare the results of this very-low incidence COVID period with the ones with a very high incidence (e.g. the following Autumn-Winter);

2. Authors should better discuss the fact that vaccination uptake was the lowest in obese individuals. This may be the major explanation of their results and, therefore requires more attention in discussion;

3. In fact, it is particularly intriguing that when correcting for vaccination status the association between obesity and hospital admission was not observed. This is in contrast with previous reports in COVID ad well ad in a more general term with previous similar conditions where the association between obesity and influenza-like or other respiratory virus infection were studied (see Luzi et al, Acta Diabetologica 2020 for a thourogh analysis of literature);

3. Besides the quoted article, more in general, pre-existing literature should be better quoted and discussed;

4. Discussion should be more focused and trimmed by 30%.

Author Response

Dear Reviewer 1,

Thank you for your review and comments. Please see the attachment.

Kind regards,

Maika

Reviewer 2 Report

Dear Authors,

I think that your work is very substantial and interesting especially for the sample size.... but in my opinion it is too confusing and the section about the results and conclusions should be better organized to make it easier to understand to the readers. 

The same applies for tables and figures.

Author Response

Dear Reviewer 2,

Thank you for your review and comments. Please see the attachment.

Kind regards,

Maika

Round 2

Reviewer 2 Report

Dera Authors , 

I noticed your corrections that you have made after my suggestions. Moreover, I have to suggest you to modify again both the figure 2 and table 3 that are not well formatted. 

Author Response

Dear Reviewer,

Thank you for your review and your suggestions.

We can see Figure 2 correctly from the latest file, and there is no Table 3 in our manuscript. The editor also confirmed Figure 2 shown correctly. However, we see that Tables 1 and 2 may need formatting as rows are breaking across pages differently than before due to editing, and Figure 3 may need to be repositioned.

In view of the above, we amended the following:

Tables 1 and 2 are now fit into a page.  Figure 3 is now repositioned. 

Thank you again for your suggestions.